# Selective Degradation and Inhibition of SARS-CoV-2 3CL^pro^ by MMP14 Reveals a Novel Strategy for COVID-19 Therapeutics

**DOI:** 10.3390/ijms26199401

**Published:** 2025-09-26

**Authors:** Hyun Lee, Yunjeong Hwang, Elizabeth J. Mulder, Yuri Song, Calista Choi, Lijun Rong, Dimitri T. Azar, Kyu-Yeon Han

**Affiliations:** 1Department of Pharmaceutical Science, College of Pharmacy & Biophysics Core, the Research Resource Center, University of Illinois Chicago, Chicago, IL 60607, USA; danielhl@uic.edu; 2Biophysics Core, the Research Resource Center, University of Illinois Chicago, Chicago, IL 60607, USA; ejmulder@uic.edu (E.J.M.); ysong211@uic.edu (Y.S.); cchoi46@uic.edu (C.C.); 3Department of Ophthalmology and Visual Sciences, Illinois Eye and Ear Infirmary, College of Medicine, University of Illinois Chicago, Chicago, IL 60612, USA; yjhwang1@uic.edu; 4Department of Microbiology and Immunology, College of Medicine, University of Illinois Chicago, Chicago, IL 60612, USA; lijun@uic.edu

**Keywords:** SARS-CoV-2, MMP14, 3CL^pro^, enzyme inhibition

## Abstract

Novel therapies to treat infection by severe acute respiratory syndrome coronavirus 2 (SARS-CoV-2), the cause of respiratory coronavirus disease 2019 (COVID-19), would be of great clinical value to combat the current and future pandemics. Two viral proteases, papain-like protease (PL^pro^) and the main protease 3-chymotrypsin-like protease (3CL^pro^), are vital in processing the SARS-CoV-2 polyproteins (pp1a and pp1ab) and in releasing 16 nonstructural proteins, making them attractive antiviral drug targets. In this study, we investigated the degradation of the SARS-CoV-2 main protease 3CL^pro^ by matrix metalloproteinase-14 (MMP14). MMP14 is known to recognize over 10 distinct substrate cleavage sequences. Through sequence analysis, we identified 17 and 10 putative MMP14 cleavage motifs within the SARS-CoV-2 3CL^pro^ and PL^pro^ proteases, respectively. Despite the presence of potential sites in both proteins, our in vitro proteolysis assays demonstrated that MMP14 selectively binds to and degrades 3CL^pro^, but not PL^pro^. This selective proteolysis by MMP14 results in the complete loss of 3CL^pro^ enzymatic activity. In addition, SARS-CoV-2 pseudovirus replication was inhibited in 293 T cells when either full-length MMP14 or its catalytic domain (cat-MMP14) were overexpressed, presumably due to 3CL^pro^ degradation by MMP14. Finally, to prevent MMP14 from degrading off-target proteins, we propose a new recombinant pro-PL-MMP14 construct that can be activated only by another SARS-CoV-2 protease, PL^pro^. These findings could open the potential of an alternative therapeutic strategy against SARS-CoV-2 infection.

## 1. Introduction

Severe acute respiratory syndrome coronavirus (SARS-CoV) emerged in late 2002 from China and Hong Kong and rapidly infected ~8000 people worldwide, with a fatality rate of approximately 10% [1]. It was contained through epidemiological and public quarantine measures within several months. Eighteen years later, another human coronavirus (SARS-CoV-2) was reported in Wuhan, China, in late December 2019 and quickly spread to more than 200 countries. SARS-CoV-2 is more transmissible/contagious than the previous SARS-CoV based on its higher basic reproduction number (R_0_), which ranges from 1.4 to 6.49, according to different reports [2,3]. The rapid spread of SARS-CoV-2 around the world is believed to have been facilitated by transmission from asymptomatic persons to others [4]. The COVID-19 vaccine could possibly end this current pandemic. However, SARS-CoV-2 has been predicted to possibly behave similarly to the influenza virus, re-emerging every year in slightly different forms. More importantly, the SARS-CoV-2 virus keeps making various mutants with higher infection rates than previous ones, threatening public health. Currently, no established treatments are available for eliminating SARS-CoV or SARS-CoV-2.

Coronaviruses belong to the Coronaviridae family in the Nidovirales order [5], and SARS-CoV-2 belongs to the betacoronavirus genus, along with SARS-CoV and Middle East Respiratory Syndrome Coronavirus (MERS-CoV). Coronaviruses contain a positive-sense single-stranded RNA ranging from 26 to 32 kb in length [6]. The SARS-CoV-2 genome shares around 70–80% sequence identity with that of the previous SARS-CoV and uses the same ACE2 protein as an entry receptor into the host [7,8], while DPP4 acts as a receptor for MERS-CoV [9,10,11]. The SARS-CoV-2 gene encodes polyproteins (pp1a and pp1ab) to be processed by two viral proteases, papain-like protease (PL^pro^) and 3-chymotrypsin-like protease (3CL^pro^), which results in the release of 16 nonstructural proteins (NSPs) (Figure 1A). Both the PL^pro^ and 3CL^pro^ proteases are considered attractive drug targets due to their essentiality for viral replication. The similarity in protein sequence identities of SARS-CoV-2 PL^pro^ and 3CL^pro^ to those of the original SARS-CoV are 82.9% and 96.1%, respectively [12]. Figure 1B compares the structures of the SARS-CoV and -2 3CL^pro^, highlighting the catalytic dyad, His41 and Cys145, and four loops crucial for substrate recognition. Residues that differ between SARS-CoV and SARS-CoV-2 3CL^pro^ are shown in brown and blue, respectively, and most of them are located in other areas, except one residue (Ala46) in loop 4 [13]. As expected, these two structures are very similar. SARS-CoV 3CL^pro^ is responsible for cleaving 11 locations in its polyproteins (pink arrows in Figure 1A) and releasing 13 NSPs, including its viral polymerase and helicase, two other known important drug targets.

Various therapeutic strategies have been explored to inhibit SARS-CoV-2 replication and pathogenesis. Among these, drug repurposing has emerged as a rapid and pragmatic approach for identifying potential antiviral agents. Approved therapeutics such as remdesivir and molnupiravir, which target the viral RNA polymerase, have shown varying degrees of clinical efficacy [14,15]. Nystatin, a polyene macrolide antifungal, has been proposed as a non-systemic antiviral candidate due to its strong affinity for cholesterol-rich membrane domains. It replication of both the Wuhan and B.1.1.7 (UK) variants in Vero E6 cells by disrupting lipid rafts essential for viral entry and gastrointestinal persistence [16]. Beyond repurposed agents, structure-based design has enabled the development of selective inhibitors of viral proteases, including 3CL^pro^ and PL^pro^. Compounds such as nirmatrelvir (Paxlovid), PF-00835231, and GC-376 have demonstrated potent activity in biochemical and cell-based assays [17,18,19]. Other therapeutic modalities include host-directed approaches (e.g.,TMPRSS2 inhibitors [20]), monoclonal antibodies [21], and proteolysis-targeting chimeras (PROTACs), which are engineered to degrade viral or host proteins [22]. Despite these advances, selectively targeting viral proteases remains challenging due to specificity concerns and the risk of resistance development.

Membrane-bound matrix metalloproteinase-14 (MMP14, MT1-MMP) is one of the most well-studied members of the metalloproteinase family, both in vivo and in vitro, and in both normal physiological and disease models. MMP14 becomes active when its pro-domain is cleaved by furin (Figure 1C), and the MMP14 catalytic domain (cat-MMP14) X-ray structure revealed that it can bind two calcium and two zinc ions, shown in green and grey, respectively, in Figure 1D. MMP14 has a variety of roles in processes that involve angiogenesis, including wound healing, inflammation, and cancer invasion and metastasis [23,24]. Classically, MMP14 is regarded as a degrader of the extracellular matrix (ECM) that is responsible for remodeling matrix proteins [25]. Recently, MMP14 has also been recognized as a regulator of chemokines, cytokines, and growth factors [26,27], but cat-MMP14 has been shown to have similar properties to the full-length MMP14 with the exception of matrix degradation [28]. MMP14 also plays a well-established role in activating pro-MMP2 through proteolytic cleavage, a function commonly used as a marker of its catalytic activity [29]. Our previous research demonstrated that MMP14 selectively and uniquely cleaves nonfunctional vascular endothelial growth factor receptor 1 (VEGFR1), which acts as a decoy receptor to mostly functional VEGFR2. The proteolysis of VEGFR1 by MMP14 increases the availability of VEGFA to bind to VEGFR2, which promotes VEGFA/VEGFR2-mediated corneal neovascularization [30,31].

We hypothesize that MMP14 can selectively degrade 3CL^pro^, thereby inhibiting viral replication. MMP14 has at least eight different reported cleavage sequences. We performed a search on these eight cleavage sequences and found 17 potential MMP14 cleavage sites on SARS-CoV-2 3CL^pro^ (Figure 2A). Next, we studied the proteolytic activities of MMP14 on two essential viral proteases, SARS-CoV-2 3CL^pro^ and PL^pro^. The results presented herein show that MMP14 effectively cleaves SARS-CoV 3CL^pro^, whereas it does not cleave PL^pro^. In addition, the MMP14 proteolysis of 3CL^pro^ was inhibited by a pan-MMP inhibitor, GM6001, and 3CL^pro^ enzyme activity was gradually eliminated upon cleavage by MMP14. This interaction between MMP14 and 3CL^pro^ was also confirmed via a direct binding analysis. Finally, a SARS-CoV-2 pseudovirus replication assay further confirmed this activity and guided the design of a recombinant construct that is specifically activated by SARS-CoV-2 PL^pro^, enabling selective action in infected cells.

## 2. Results

### 2.1. MMP14 Cleaved 3CL^pro^ but Not PL^pro^

To investigate the proteolytic effect of MMP14 on two essential SARS-CoV-2 proteases, PL^pro^ and 3CL^pro^, we performed in vitro proteolysis assays. After incubating the catalytic domain of MMP14 with SARS-CoV-2 PL^pro^ and 3CL^pro^, based on a signal intensity analysis using densitometry (Odyssey, Li-cor), we observed that the 3CL^pro^ protein was completely degraded by the catalytic domain of MMP14 (cat-MMP14), while PL^pro^ was not cleaved by cat-MMP14 (Figure 2A). A newly generated smaller fragment protein band was observed between 17 kDa and 10 kDa. This band was distinguishable from cat-MMP14 (21 kDa) and only appeared in 3CL^pro^ and MMP14 mixture samples, but not in the PL^pro^ and MMP14 mixture (Figure 2A). To further characterize the degradation efficiency of the MMP14 against SARS-CoV-2 3CL^pro^, we performed in vitro proteolysis assays at various time points up to 2 h. Equal amounts of 3CL^pro^ and MMP14 were used in each reaction, and uniform protein loading across lanes was verified by consistent MMP14 band intensities and Coomassie Blue staining of duplicate gels. Based on densitometric analysis of the 3CL^pro^ bands, we observed that at least 75% of the intact form of 3CL^pro^ was degraded by MMP14 within the first 15 min (red box in Figure 2C), accompanied by the appearance of a smaller degradation fragment. We also observed 3CL^pro^ degradation over time via mass photometry and saw a steady decrease between about 5 and 15 min of the 3CL^pro^ dimer peak (the monomer is too small to resolve via mass photometry) (Figure 2D).

To determine the cleavage site of 3CL^pro^ by MMP14, we incubated SARS-CoV-2 3CL^pro^ with cat-MMP14 and performed Edman sequencing on the resulting 3CL^pro^ fragment. We compared the sequence of amino acids from the Edman degradation result (Figure 3) to the 3CL^pro^ sequence and potential cleavage sites (Figure 2A) and identified PTG|VHAGT as the cleavage site. The predicted MW of this fragment (14.85 kDa) matches the observed MW on SDS-PAGE (Figure 2B,C), further confirming that MMP14 cleaves 3CL^pro^ between G170 and V171 which is located in the loop 2 (Figure 2B).

To confirm that the cleavage of 3CL^pro^ is dependent on the catalytic activity of MMP14, proteolytic assays were repeated in the absence and presence of a pan-MMP inhibitor GM6001. The results revealed that GM6001 completely inhibited 3CL^pro^ degradation by MMP14 (Figure 4A). Next, we investigated whether some other MMPs have similar proteolytic effects on either SARS-CoV-2 3CL^pro^ or PL^pro^. Four additional MMPs, MMP-2, -7, -9, and -13, were tested with PL^pro^ or 3CL^pro^. Interestingly, these four MMPs exhibited no or very minor cleavage activities on either PL^pro^ or 3CL^pro^ (Figure 4B). Taken together, these results indicate that MMP14 selectively degrades the crucial viral main protease 3CL^pro^ in vitro and that the catalytic enzyme activity of MMP14 is essential for 3CL^pro^ degradation.

### 2.2. MMP14 Directly Binds to 3CL^pro^

We observed that cat-MMP14 could degrade SARS-CoV-2 3CL^pro^, and presumably, these two proteins should directly bind first for MMP14 to degrade 3CL^pro^. Hence, we examined whether they can directly bind to each other using surface plasmon resonance (SPR). The SPR technique can monitor real-time binding events by measuring angle changes in reflected light caused by refractive index changes upon binding (Figure 5A). The binding kinetics of the cat-MMP14 to SARS-CoV-2 3CL^pro^ were investigated by measuring how quickly they bind (*k*_a_), how tightly they bind (*K*_D_) at equilibrium, and how quickly they dissociate (*k*_d_). 3CL^pro^ clearly showed binding to the immobilized MMP14, with a binding affinity (*K*_D_) of 3.21 ± 0.95 (μM) (Figure 5B). In contrast, SARS-CoV-2 PL^pro^ did not bind to the immobilized MMP14 (Figure 5C). 3CL^pro^ sensorgrams were fitted with a 1:1 Langmuir kinetic model, as shown by the black dotted lines in Figure 5B. The association rate (*k*_a_) and dissociation rate (*k*_d_) constants for 3CL^pro^ binding to MMP14 were determined to be 3.86 × 10^2^ M^−1^ s^−1^ and 1.20 × 10^−3^ s^−1^, indicating that 3CL^pro^ binds slowly to MMP14 and dissociates very slowly. Even though the sensorgrams did not reach a plateau after 180 s of sample injections, steady-state affinity fitting curves for both 3CL^pro^ and PL^pro^ are shown simply for comparison purposes (Figure 5D). Since the molecular weight of the PL^pro^ is slightly bigger than that of 3CL^pro^, the binding response of the PL^pro^ should be higher than that of 3CL^pro^, but we observed the opposite (Figure 5D). These SPR binding analysis results confirmed again that MMP14 interacts only with 3CL^pro^ and not PL^pro^.

### 2.3. Elimination of 3CL^pro^ Proteolytic Activity by MMP14

To determine whether the enzyme activity of 3CL^pro^ is inhibited upon degradation by MMP14, we performed a 3CL^pro^ enzymatic assay using a fluorogenic 3CL^pro^ substrate 5-FAM-TSATLQSGFRK-(QXL520)-NH_2_. This substrate contains a native cleavage site between nsp4 and nsp5 (3CL^pro^) (Figure 1A). 3CL^pro^ cleaves the substrate between a glutamine (Q) molecule and a serine (S) one, and a fluorescence signal is generated from the fluorophore FAM upon this cleavage by 3CL^pro^ because the QXL quencher is eliminated. To confirm the suitability of this substrate for further assays, we first observed the 3CL^pro^ activity with this substrate in a dose-dependent manner (Figure 6A). Then, since MMP14 cleaves more than 10 reported sequences, we needed to confirm that MMP14 does not cleave our 3CL^pro^ substrate. To achieve this, we performed the same experiment with a series of increasing concentrations of the catalytic domain of MMP14 without any 3CL^pro^ present. MMP14 was unable to cleave the 3CL^pro^ substrate, and only very minor MMP14 activity was observed at a very high concentration (Figure 6B). Subsequently, we observed that MMP14 inhibited the 3CL^pro^ enzyme activity in a dose-dependent manner (Figure 6C), and we can attribute this effect to the degradation of 3CL^pro^ by MMP14. These results suggest that MMP14 may have an inhibitory effect on 3CL^pro^ enzyme activity in SARS-CoV-2-infected cells, and this finding may lead to a new proof-of-concept for the development of drugs that prohibit SARS-CoV-2 viral replication by eliminating its main protease 3CL^pro^.

### 2.4. Inhibition of SARS-CoV-2 Pseudovirus Replication

The next step was to investigate whether the expression of MMP14 inhibits the replication of SARS-CoV-2 in cells. We prepared 293T cells transfected with an empty vector as a control and with various amounts of full-length MMP14 (0.5, 5, and 10 µg), followed by infection with a SARS-CoV-2 pseudovirus that has a luminescence gene inserted into its sequences. The luminescence signal of the 293T cells transfected with full-length MMP14 decreased in a dose-dependent manner in comparison to that of the control 293T cells (Figure 7A), indicating that the replication of the SARS-CoV-2 pseudovirus was inhibited. To determine whether the replication of the SARS-CoV-2 pseudovirus should be inhibited with cat-MMP14, the same experiment was repeated with cat-MMP14-expressed cells. The percent inhibitions of the SARS-CoV-2 pseudovirus by cat-MMP14 were similar to those of the full-length MMP14, reaching over 50% inhibition with 10 µg transfection of both (Figure 7B). Interestingly, we observed ~20% inhibition of the pseudovirus replication at 0.5 µg transfection with cat-MMP14, whereas very low to no inhibition was observed from the same 0.5 µg transfection with full-length MMP14. Next, a gelatin zymogram analysis was performed to determine the activation of pro-MMP2 to active MMP2 (act-MMP2) by membrane-anchored MMP14 (Figure 7C). We observed that the overexpression of the empty vector or cat-MMP14 plasmid did not show the cleaved form of MMP2 (act-MMP2). However, the conditioned media from the overexpression of full-length MMP14 in 293T cells contained the act-MMP2 cleaved from pro-MMP2. This result indicates that the catalytic domain of MMP14 may be better able to inhibit the replication of SARS-CoV-2.

### 2.5. A Newly Engineered Pro-PL-MMP14 Was Activated by SARS-CoV-2 PL^pro^

MMP14 is involved in several physiological processes with various cleavage sites, raising concerns of unwanted adverse effects if we are to use it for SARS-CoV-2 treatment. Hence, we need a good strategy to avoid this possibility. The pro-MMP14 is inactive, and furin transiently binds and cleaves the pro-domain sequence to yield an active form of MMP14 (act-MMP14). We have designed a new MMP14 construct, pro-PL-MMP14, with a SARS-CoV-2 PL^pro^ cleavage site (LKGGA) between the pro- and active domains of MMP14. Thus, the MMP14 will not become active without exposure to SARS-CoV-2 PL^pro^. The construct also has a C-terminal His_6_-tag (for purification) and Arg108 mutated to Pro to avoid cleavage by furin (and thus, activation in the absence of PL^pro^) (Figure 8A). We were able to purify this pro-PL-MMP14 protein and verify this purification using SDS-PAGE and Western blotting analyses against two antibodies: anti-His and anti-MMP14. To investigate the activation of pro-PL-MMP14 by SARS-CoV-2 PL^pro^, we also purified the SARS-CoV-2 PL^pro^ enzyme. Act-MMP14 (cleaved form) was produced in the presence of PL^pro^ (Figure 8B, blue box). Act-MMP2 was generated only when both pro-PL-MMP14 and PL^pro^ were present, indicating activation of pro-PL-MMP14 by PL^pro^ (Figure 8C). Therefore, this modified recombinant pro-PL-MMP14 will be activated exclusively in SARS-CoV-infected cells when PL^pro^ is available, thereby avoiding potential adverse effects in uninfected cells. Taken together, our results suggest the potential of employing cat-MMP14 as a novel therapeutic approach to supplement current treatments aimed at inhibiting SARS-CoV-2 replication.

## 3. Discussion

Even though various SARS-CoV-2 vaccines are currently available for SARS-CoV-2 infections, therapeutics are still needed for people who cannot be vaccinated or who lack immunity development. In this study, we demonstrated a new function of the catalytic domain of MMP14 against an essential viral protease, 3CL^pro^ of SARS-CoV-2. The catalytic domain of MMP14 uniquely bound to SARS-CoV-2 3CL^pro^ and consequently degraded 3CL^pro^. However, this was not the case for SARS-CoV-2 PL^pro^, suggesting that this proteolytic activity of MMP14 is specific to 3CL^pro^. Based on these observations, we hypothesized that MMP14 may directly bind not to PL^pro^ but to 3CL^pro^ to degrade 3CL^pro^. An SPR analysis of various 3CL^pro^ concentrations clearly showed 3CL^pro^ binding to the immobilized MMP14, whereas PL^pro^ did not (Figure 4). Interestingly, four other MMPs (i.e., MMP-2, -7, -9, and -13) could not proteolyze either viral protease. Furthermore, the degradation of 3CL^pro^ by MMP14 led to the inhibition of 3CL^pro^ enzyme activity. These results support the possibility of the catalytic domain of MMP14 inhibiting SARS-CoV-2 replication by recognizing 3CL^pro^ as its substrate and thus blocking the release of 13 viral proteins, including the RNA polymerase crucial for viral replication in SARS-CoV-2-infected cells.

Over almost two decades, researchers have been trying to develop inhibitors against essential viral drug targets such as 3CL^pro^, PL^pro^, polymerase, and helicase. However, various obstacles have prevented the development of such inhibitors, and we believe that using host proteases to eliminate essential viral proteases could be another method of preventing viral replication. MMP14 is responsible not only for the degradation of ECM components and activation of pro-MMP2 but also the regulation of non-ECM proteins such as cytokines, growth factors, and receptors [31,32,33,34,35,36]. Based on our extensive experience with MMP14, we searched all known MMP14 cleavage sites in SARS-CoV-2 3CL^pro^ and PL^pro^ and found seventeen and ten potential cleavage sites on 3CL^pro^ and PL^pro^, respectively. In our in vitro proteolysis, we observed a very faint fragment from 3CL^pro^ degradation. This result may suggest that cat-MMP14 cleaves 3CL^pro^ at multiple sites, although only one major cleavage fragment was detected. It is also important to consider the cellular localization of MMP14. As a membrane-bound protease, MMP14 is generally thought to be spatially restricted to the cell surface, which could limit its access to cytosolic viral proteins such as 3CL^pro^. Furthermore, MMP14 is synthesized as an inactive pro-enzyme that requires activation by furin in the Trans-Golgi network. Therefore, for MMP14 to effectively degrade 3CL^pro^ within infected cells, it would need to transition from its conventional membrane-associated role to a cytosolic context, either through alternative trafficking or the use of a truncated catalytic form lacking the transmembrane domain [37,38]. In either case, the active MMP14 would then be stored in late and recycled endosomes [39]. Despite being spatially confined within the cell, MMP14 has been reported to retain enzyme activity in the cytosol and to cleave pericentrin within cells [40]. SARS-CoV-2 PL^pro^ and 3CL^pro^ also act in the cytosol to process viral polyproteins and generate a functional replicase complex [41,42]. This discovery means that MMP14 and 3CL^pro^ may interact within infected cells, despite the conventional understanding of MMP14 spatial restrictions. In our experiment using a SARS-CoV-2 pseudovirus, both types of MMP14 (catalytic domain and full-length) effectively inhibited virus replication. In particular, a low concentration of the catalytic domain of MMP14 showed a better efficiency than full-length MMP14 at the same concentration (Figure 7A,B). We observed that cat-MMP14, which lacks the transmembrane domain, more effectively inhibited pseudovirus replication than the full-length MMP14. This suggests that when not membrane-anchored, MMP14 may interact with SARS-CoV-2 3CL^pro^ within the cytoplasm, and that its catalytic domain plays a key role in this antiviral activity. Although the mechanism by which cat-MMP14 gains cytoplasmic access remains to be elucidated, recent studies have identified exosomes as natural carriers capable of delivering functional proteins into recipient cells via membrane fusion or endosomal escape. While further investigation is needed, exploring exosome-mediated delivery of cat-MMP14 could help elucidate its intracellular trafficking and enhance its therapeutic potential. It is important to note that our findings are based on pseudovirus-based assays, which model key aspects of viral entry and replication. Therefore, further studies using live SARS-CoV-2 are necessary to confirm the antiviral efficacy of cat-MMP14 in the context of authentic viral infection. Collectively, these findings support cat-MMP14 as a promising candidate for development as an antiviral agent against SARS-CoV-2.

## 4. Materials and Methods

### 4.1. Expression and Purification of SARS-CoV-2 PL^pro^ and 3CL^pro^ and Human Pro-PL-MMP14

The full-length SARS-CoV-2 3CL^pro^ gene was codon-optimized, synthesized, and cloned into a pGEX6p-1 expression vector with HRV-cleavable his6-tag at the C-terminus. The overall overexpression and purification procedures were performed similarly to those described in [43]. In short, 2 L of BL21(DE3) cells was grown to an OD_600_ of 0.6 at 37 °C in LB medium, followed by 0.5 mM IPTG induction for 15 h at 25 °C. The harvested cells were lysed through sonication and purified via a two-step (1 mL of HisTrap affinity-16/60 Superdex 75 SEC) purification protocol using AKTA Pure FPLC (Cytiva, Marlborough, MA, USA). The his-tag was cleaved by HRV protease, producing native 3CL^pro^ proteins. The SARS-CoV-2 PL^pro^ gene was also codon-optimized, synthesized, and cloned into pET11a expression vectors with TEV-cleavable his6-tag at the N-terminus. Then, 2 L of BL21(DE3) cells was grown to an OD_600_ of 0.6 at 37 °C in LB medium, followed by 0.5 mM IPTG induction for 16 h at 18 °C. The harvested cells were lysed through sonication and purified via a two-step (1 mL of HisTrap affinity-16/60 Superdex 75 SEC) purification protocol followed by his-tag cleavage using TEV protease. Human R108P-PL-MMP14(90-287)-His was codon-optimized, synthesized, and cloned into pET11a expression vectors with a his6-tag at the C-terminus (GenScript, Piscataway, NJ, USA). Then, 2 L of BL21(DE3) cells (Agilent, Santa Clara, CA, USA) was grown to an OD_600_ of 2.0 in SuperLB medium (6 mg/mL of sodium phosphate, 3 mg/mL of potassium phosphate, 20 mg/mL of tryptone, 5 mg/mL of yeast extract, and 5 mg/mL of sodium chloride, pH 7.2), followed by auto-induction for 18 h at 18 °C. The harvested cells were mixed into a lysis buffer (50 mM Tris pH 7.5, 500 mM NaCl, 5 mM bME, 10 mg/mL lysozyme, 1% Triton X-100, and DNase), lysed through sonication, and centrifuged into pellet-insoluble proteins. The pellets were dissolved in 6 M urea buffer (50 mM Tris, 500 mM NaCl, 5 mM bME, and 6 M Urea, pH 8.5) to denature the proteins and purified with 1 or 5 mL of HisTrapHP (Cytiva #17524701) affinity purification (eluted with 500 mM imidazole) on AKTA Pure FPLC (Cytiva). Elution fractions were dialyzed against a gel filtration buffer (50 mM Tris, 500 mM NaCl, 150 mM bME, and 1 M Urea, pH 8.0) overnight and purified further on a HiLoad 16/60 or 26/60 Superdex 200 pg size exclusion column (Cytiva #28989336) equilibrated with a gel filtration buffer. The protein solution was then concentrated to 0.1–0.2 mg/mL with an amicon centrifugal filter (MWCO 3 kDa) (Millipore #UFC9003), buffer-exchanged with Zeba desalting spin columns (Thermo Scientific, Waltham, MA, USA ) to the final storage buffer (50 mM Tris, 150 mM NaCl, 0.1 mM ZnCl_2_, 5 mM CaCl_2_, pH 7.5), and incubated on ice for 2 h to induce refolding before flash-freezing and storage at −80 °C. Purity was confirmed via SDS-PAGE.

### 4.2. Mass Photometry

The potential degradation and binding to MMP14 of 3CL^pro^ and PL^pro^ was monitored with a TwoMP™ mass photometer (Refeyn Ltd., Oxford, UK). The TwoMP™ mass photometer was set up using ready-to-use sample carrier slides (Refeyn, Oxford, United Kingdom). The 6-well gasket was positioned on the slide with the alignment kit to ensure accurate well placement. β-amylase (BAM; 56, 112, and 224 kDa) and thyroglobulin (TG; 335 and 669 kDa) were used as molecular weight standards when calibrating the instrument. All calibrants were prepared in sterile PBS (20 mM phosphate, pH 7.4, 137 mM NaCl, and 2.7 mM KCl). Each well received 10 µL of PBS, followed by immediate mixing of a 20 nM protein, yielding a final concentration of 10 nM. Data was acquired for 60 s at room temperature using AcquireMP 2025 R1.2 software (Refeyn, Oxford, United Kingdom). The resulting calibration curve showed excellent linearity (R^2^ = 1.000) with a maximum mass error of 0.8%. PBS alone was used as a negative control to confirm the background signal before sample measurement. The stock concentrations of catalytic MMP14 (EMD Millipore, Burlington, MA, USA), purified 3CL^pro^, and PL^pro^ were 0.2 mg/mL, 2.14 mg/mL, and 2.51 mg/mL, respectively. All proteins were diluted in PBS to a final concentration of 10 nM before MP data collection. For the interaction assays, a mixture of 5 nM MMP14 and 5 nM 3CL^pro^ was measured at 0, 6, 14, 19, 24, and 38 min following incubation on ice. A separate mixture containing 2.5 nM MMP14 and 10 nM 3CL^pro^ was assessed at 0, 5, 10, 15, 20, 25, and 30 min after ice incubation. Additionally, a mixture of 5 nM MMP14 and 5 nM PL^pro^ was measured at 0, 5, 13, 18, and 32 min under the same conditions. For each time point, the protein mixture was combined with 10 µL of PBS directly on the sample carrier slide, and mass photometry data were collected for 60 s. All data were processed and analyzed using DiscoverMP 2025 R1 software (Refeyn). Single protein controls were measured in triplicate for each condition, while protein mixture samples were measured in duplicates on two different days.

### 4.3. Edman Degradation

An in vitro proteolysis assay was performed to investigate the degradation of 3CL^pro^ by MMP14. SARS-CoV-2 3CL^pro^ and cat-MMP14 were co-incubated at 37 °C for 3 h. Following cleavage, the reaction mixture (containing 1.2 µg of MMP14 and 2 µg of 3CL^pro^) was resolved via SDS-PAGE and transferred onto a polyvinylidene difluoride (PVDF) membrane. Protein bands of interest were excised in-house and submitted to the Protein Facility at Iowa State University for N-terminal sequencing via capillary-based Edman degradation. This analysis was conducted to identify the MMP14 cleavage sites. During Edman degradation, N-terminal amino acids were sequentially released as phenylthiohydantoin (PTH) derivatives and identified via HPLC. The resulting sequences were compared to the known 3CL^pro^ sequence to determine the cleavage position.

### 4.4. In Vitro Proteolysis

To investigate the proteolytic activity of MMPs on two viral proteases, SARS-CoV-2 PL^pro^ and 3CL^pro^, 1 µg of PL^pro^ or 3CL^pro^ was incubated with or without 0.6 µg of MMPs (MMP2, 3, 7, 9, and 13 from Sigma-Aldrich, St. Louis, MO, USA; MMP14 from Calbiochem, Billerica, MA, USA) in 30 μL of MMP reaction buffer (50 mM Tris-HCl, pH 7.8; 0.2 M NaCl; and 5 mM CaCl_2_) including 0.02% Brij 35 (*w/v*) (Invitrogen, Carlsbad, CA, USA) and then incubated at 37 °C for 3 h. The pan-MMP inhibitor GM6001 (Calbiochem) was used for inhibition of the MMP14 enzymatic activity. The samples were subjected to sodium dodecyl sulfate (SDS)-polyacrylamide gel electrophoresis (PAGE) (Bio-Rad, Hercules, CA, USA) for separation, and the gels were stained with AcquaStain solution (Bulldog Bio, Portsmouth, NH, USA). The proteolysis assays were repeated on a different day to confirm the results.

### 4.5. Surface Plasmon Resonance (SPR) Analysis

The catalytic domain of MMP14 was purchased (Calbiochem) and buffer-exchanged with PBSP (20 mM phosphate, pH7.4, 137 mM NaCl, 2.7 mM KCl, and 0.05% surfactant P20). Then, it was diluted to 25 µg/mL with 10 mM sodium acetate at pH 4.0 and immobilized on a CM5 sensor surface after being first activated by a 1-ethyl-3-(3-dimethylaminopropyl) carbodiimide hydrochloride (EDC)/N-hydroxy succinimide (NHS) mixture using a Biacore T200 (Cytiva). Ethanolamine blocking was performed next to deactivate the unoccupied surface area of the sensor chip. Flow channels 1 and 3 were used as references. Two proteins, SARS-CoV-2 3CL^pro^ and PL^pro^, were prepared at a series of increasing concentrations (0.062–20 µM for 3CL^pro^ and 0.041–10 µM for PL^pro^ at 2.5-fold dilution) in PBSP buffer and were applied to all four channels at a 30 µL/min flow rate with 180 s and 300 s of association and dissociation, respectively, at 25 °C. This process was repeated twice more to yield triplicate data for both 3CL^pro^ and PL^pro^. The data were double-referenced with a reference channel and zero concentration responses, and reference-subtracted sensorgrams were fitted with a 1:1 Langmuir kinetic model using Biacore Insight evaluation v6 software (Cytiva, Marlborough, MA, USA). The equilibrium dissociation constant (*K*_D_) was calculated from two obtained rate constants (*K*_D_ = *k*_d_/*k*_a_), and standard deviations were calculated from three repeats (*n* = 3).

### 4.6. SARS-CoV-2 3CL^pro^ Enzyme Assay

A fluorogenic substrate, 5-(FAM)-TSATLQSGFRK (QXL520)-NH_2_, was designed and synthesized through AnaSpec (Fremont, CA, USA) [43]. SARS-CoV-2 3CL^pro^ activity was measured using a continuous kinetic assay with this substrate. 3CL^pro^ enzyme was prepared in an assay buffer (50 mM Tris, pH 7.5, 0.01% Triton X-100, 0.1 mg/mL bovine serum albumin, and 2 mM DTT), and a series of increasing concentrations of 3CL^pro^ ranging from 0 to 1.25 µM (final concentration) was dispensed into a 96-well black plate (Corning, Tewksbury, MA, USA). The enzyme reaction was initiated by adding 3CL^pro^ substrate (5 µM final concentration) to optimize the 3CL^pro^ enzyme concentration. Various concentrations of the MMP14 enzyme were mixed with the 3CL^pro^ substrate (5 µM final concentration) to evaluate the proteolytic activity of MMP14 on the 3CL^pro^ substrate. To determine the inhibitory effect of MMP14 on 3CL^pro^ enzyme activity, a series of increasing concentrations of MMP14 enzyme was mixed with 0.15 µM (final concentration) of 3CL^pro^ into wells. The fluorescence intensity was monitored continuously at 490 nm excitation and 520 nm emission for 1 h using a Synergy h1 (Biotek, Winooski, VT, USA) late reader. All enzymatic assays were performed in triplicate (*n* = 3).

### 4.7. Inhibition of SARS-CoV-2 Replication Through MMP14 Overexpression

To determine whether the overexpression of MMP14 inhibits the replication of SARS-CoV-2 in cells, we performed a luciferase SARS-CoV-2 pseudovirus assay to measure the replication of the pseudovirus in 293T cells (ATCC, Manassas, VA, USA) transfected with an empty vector and MMP14 containing the plasmids. DNA from both cat-MMP14 (112-288) and the full-length MMP14 (NM_004995.4) were synthesized and subcloned in pcDNA3.4 between Xbal and Agel from GenScript (Piscataway, NJ, USA). For the assay, 293T cells were seeded into 6-well plates, followed by transfection with 0.5, 5, and 10 µg of both plasmids using polyethylenimine (PEI). After 5 h, the 293T cells transfected with cat-MMP14 and full-length MMP14 were collected and seeded onto 96-well plates at 10,000 cells/well, and 293T cells transfected with an empty vector were used as a control. After incubation at 37 °C, 5% CO_2_ for 24 h, the cells were infected with the SARS-CoV-2 pseudovirus. After another 24 h of incubation, the Nano-Glo luciferase assay substrate (Promega, Madison, WI, USA) was added and mixed for 5 min, and the luminescence signal of samples was measured and normalized using the luminescence signal of the control. All assays were performed in triplicate (*n* = 3).

### 4.8. Gelatin Zymogram Analysis

Equal volumes (20 μL) of conditioned media from the overexpression of the empty vector, cat-MMP14, or full-length MMP14 in 293T cells were collected in a 1.5 mL tube and then mixed with 5 μL of reducing agent free-4X SDS loading dye (Invitrogen, Grand Island, NY, USA). The samples were electrophoresed on a 10% zymogram gelatin gel (Invitrogen), followed by incubation for 1 h at room temperature. The zymogram gel was rinsed with 50 mL of deionized water twice for 30 min and then incubated with 50 mL of denaturing buffer (Invitrogen) for 2 h to remove the SDS from the gel. To induce the degradation of gelatin in the gel, the denaturing buffer was replaced with 50 mL of a developing buffer (Invitrogen). On the next day, the gel was stained with Coomassie brilliant blue R-250 (Bio-Rad), and halo images were taken using the ChemiDOC image system (Bio-Rad).

## 5. Conclusions

In summary, we explored the potential role of MMP14 as a degrader of SARS-CoV-2 3CL^pro^. First, a simple proteolysis assay of MMP14 showed cleaving of 3CL^pro^ but not PL^pro^. Next, a direct binding analysis using SPR confirmed that MMP14 was able to selectively bind to 3CL^pro^. Subsequently, MMP14 inhibited the replication of SARS-CoV-2 upon degradation of 3CL^pro^ by MMP14. We then developed a new recombinant pro-PL-MMP14 construct that can be activated only by another essential SARS-CoV-2 protease PL^pro^. This pro-PL-MMP14 does not inherently possess proteolytic activity, but it will be activated specifically by PL^pro^ to become active and degrade 3CL^pro^ to inhibit SARS-CoV-2 replication only in infected cells. These results open the door for a novel and alternative strategy for developing therapeutics using MMP14 to combat COVID-19.

## Figures and Tables

**Figure 1 ijms-26-09401-f001:**
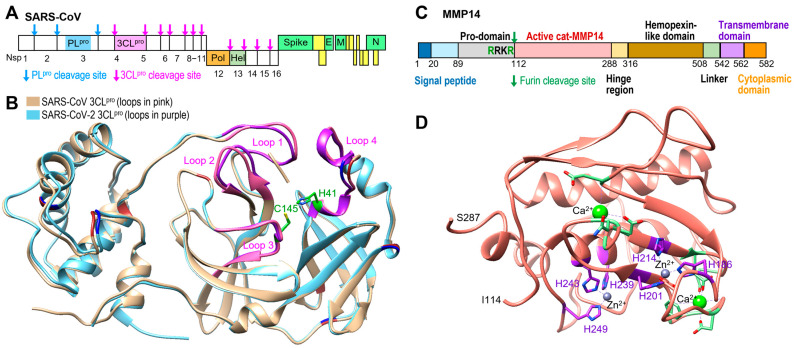
Comparison of 3CL^pro^ for the previous SARS-CoV and SARS-CoV-2. (**A**) Schematic of SARS-CoV polyproteins with two protease cleavage sites. (**B**) Overlaid structures of the previous SARS-CoV 3CL^pro^ (PDB code: 2HOB) and new SARS-CoV-2 3CL^pro^ (PDB code: 6M2Q) containing the catalytic dyad, His41 and Cys145 (green), and four loops (pink and purple) that play crucial roles in substrate recognition and enzyme activity. (**C**) Schematic of human MMP14 and furin cleavage site. (**D**) Structure of catalytic domain of MMP14 (112−290) (PDB code: 3MA2).

**Figure 2 ijms-26-09401-f002:**
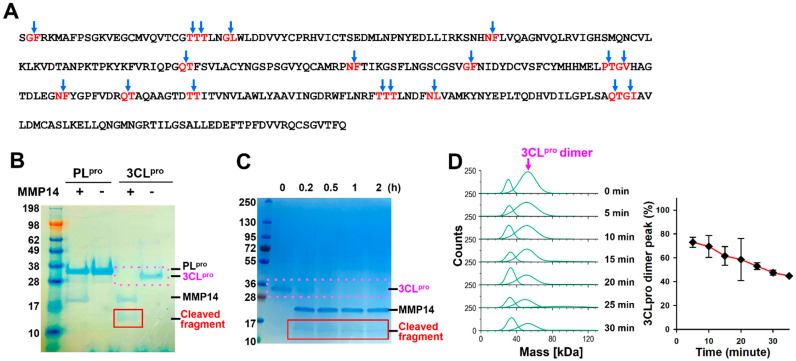
Proteolytic activity of MMP14 on SARS-CoV-2 3CL^pro^. (**A**) Protein sequence of SARS-CoV-2 3CL^pro^ and 17 potential cleavage sites for MMP14. Previously reported MMP14 recognition motifs are indicated in red, and putative cleavage sites are highlighted in blue arrow. (**B**) MMP14 cleaved SARS-CoV-2 3CL^pro^ but not PL^pro^. (**C**) Time dependence of SARS-CoV-2 3CL^pro^ cleavage by MMP14. (**D**) Mass photometry measurements of SARS-CoV-2 3CL^pro^ cleavage by MMP14 over time. Quantification of the 3CL^pro^ dimer peak is plotted (right). The experiments in (**B**–**D**) were performed twice on two different days.

**Figure 3 ijms-26-09401-f003:**
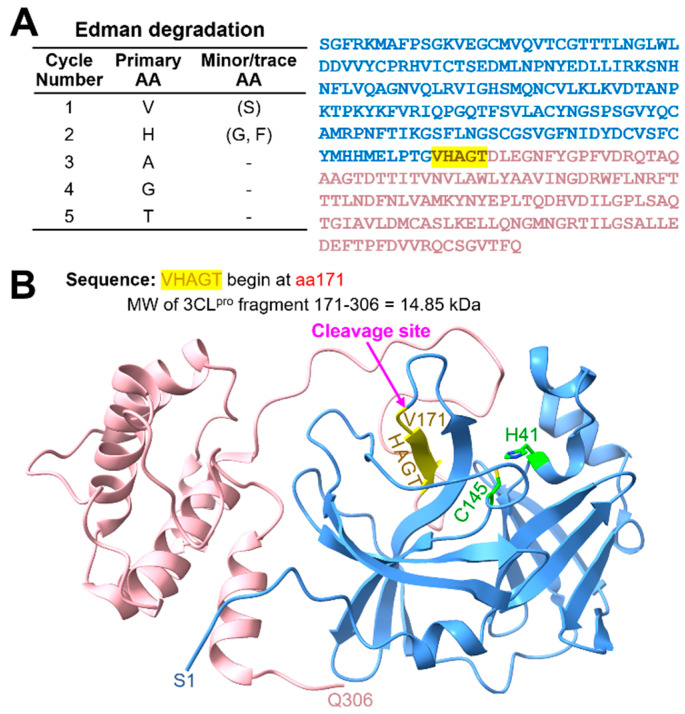
Determination of the cleavage site of 3CL^pro^ by MMP14 through Edman sequencing. (**A**) The first five N-terminal amino acids of the 3CL^pro^ fragment identified via Edman degradation. Minor/trace co-detected peaks are shown in parentheses. The cleaved 3CL^pro^ band was transferred to a PVDF membrane, and Edman degradation was used to sequentially cleave and derivatize amino acids from the N-terminus. The best match for the identified sequence “VHAGT” in the full-length 3CL^pro^ corresponds to residues 171–306 (highlighted in pale violet), while residues 1–170 are shown in blue. (**B**) Structural representation of the cleavage site within 3CL^pro^ (PDB ID: 6M2Q), located in Loop 2. The identified residues are shown in dark yellow, and the catalytic dyad is highlighted in green. The predicted molecular weight of the fragment cleaved at this position is based on this segment.

**Figure 4 ijms-26-09401-f004:**
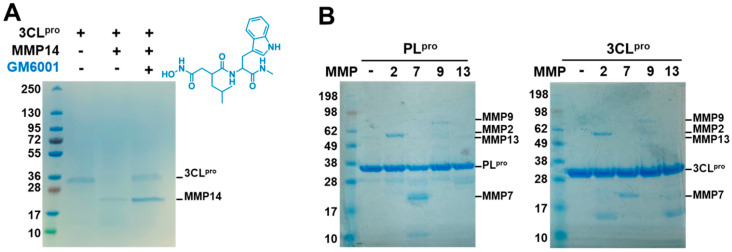
Catalytic activity of MMP14 was required for cleavage of 3CL^pro^. (**A**) Pan-MMP inhibitor GM6001 prevented MMP14 cleavage of SARS-CoV-2 3CL^pro^. The molecular weights of SARS-CoV-2 3CL^pro^, PL^pro^, and MMP14 (catalytic domain) are 36 kDa, 34 kDa, and 21 kDa, respectively. (**B**) Two MMPs, MMP-7 (19 kDa) and MMP-9 (82 kDa), could not cleave SARS-CoV-2 3CL^pro^, while the other two, MMP-2 (62 kDa) and MMP-13 (54 kDa), showed very minor cleavage. All proteolysis assays were performed twice.

**Figure 5 ijms-26-09401-f005:**
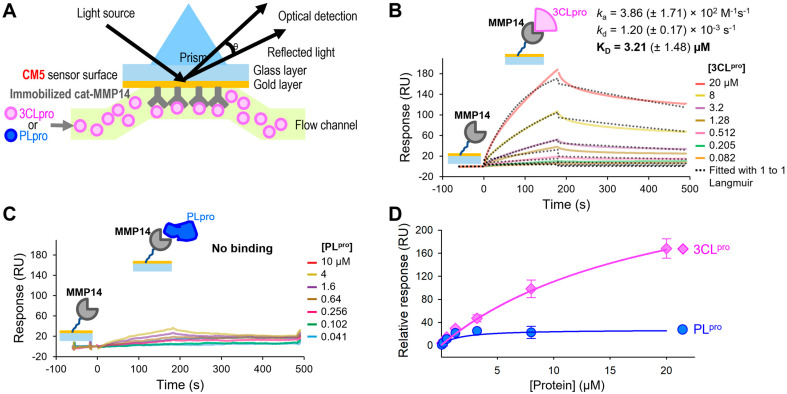
Surface plasmon resonance (SPR) binding analysis. (**A**) Sensorgrams of SARS-CoV-2 3CL^pro^ binding to immobilized MMP14 catalytic domain at a series of increasing concentrations. (**B**) Steady-state affinity fitting curves at a series of increasing concentrations of SARS-CoV-2 3CL^pro^. Sensorgrams (**C**) and the steady-state affinity fitting curves (**D**) of the SARS-CoV-2 PL^pro^ binding to immobilized MMP14 catalytic domain. The standard deviation was calculated from three independent measurements. Either SARS-CoV-2 3CL^pro^ or PL^pro^ was injected for 180 s, followed by buffer injection to monitor dissociations. Standard deviations were calculated from three repeats (*n* = 3).

**Figure 6 ijms-26-09401-f006:**
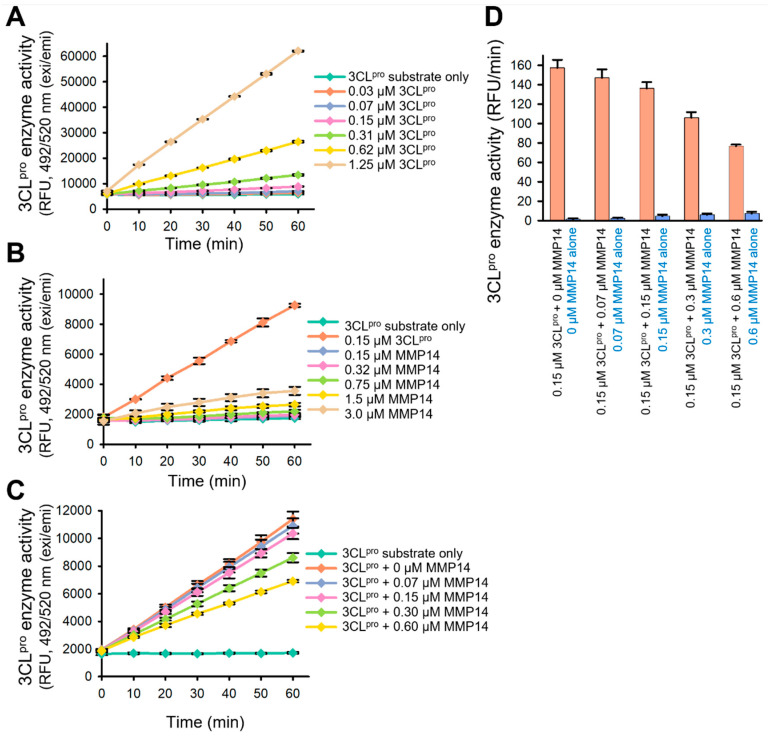
SARS-CoV-2 3CL^pro^ enzymatic assays. (**A**) Cleavage of the fluorogenic substrate by SARS-CoV-2 3CL^pro^ was both time- and concentration-dependent. (**B**) The catalytic domain of MMP14 did not cleave the fluorogenic substrate of SARS-CoV-2 3CL^pro^. (**C**) Cleavage of SARS-CoV-2 3CL^pro^ by the catalytic domain of MMP14 inhibited the enzymatic activity of SARS-CoV-2 3CL^pro^. (**D**) Bar graph comparison of 3CL^pro^ and MMP14 activities on 3CL^pro^ substrate. All enzymatic assays were performed in triplicate; error bars represent standard deviation from three independent assays (*n* = 3).

**Figure 7 ijms-26-09401-f007:**
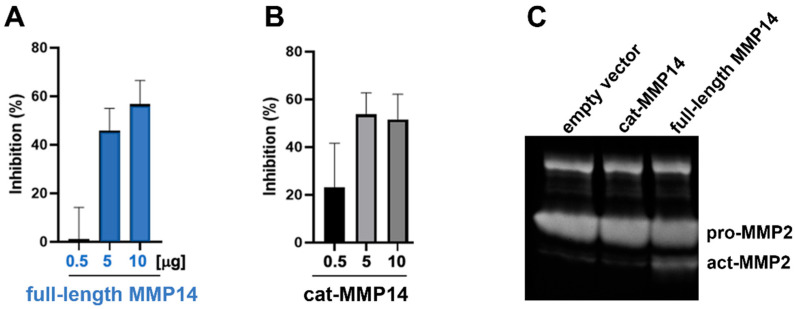
MMP14 expression inhibited SARS-CoV-2 pseudovirus replication in 293T cells. (**A**) Replication of the SARS-CoV-2 pseudovirus in 293T cells was inhibited by full-length MMP14. (**B**) Repeated replication experiment with catalytic domain of MMP14. (**C**) Gelatin zymogram analysis. Full-length MMP14 activated pro-MMP2 but cat-MMP14 expression could not activate pro-MMP2 in the gelatin zymogram analysis. Standard deviations were calculated from three repeats (*n* = 3).

**Figure 8 ijms-26-09401-f008:**
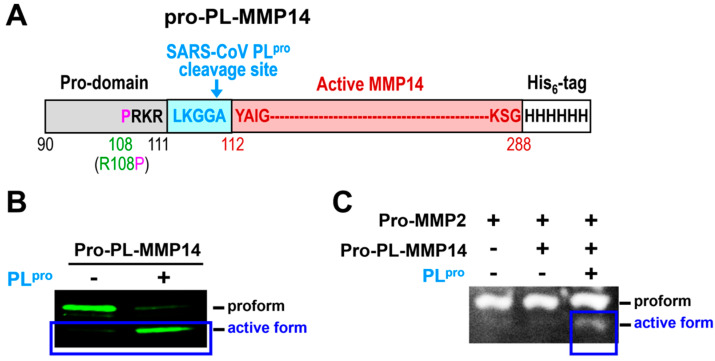
A newly engineered pro-PL-MMP14 activated by SARS-CoV-2 PL^pro^. (**A**) Schematic of the proposed construct named pro-PL-MMP14 containing a SARS-CoV-2 PLpro (another SARS-CoV protease) cleavage site (LKGGA) between the pro-domain and act-MMP14 with a C-terminal His_6_-tag, and Arg108 that has been mutated to Pro to avoid cleavage by furin. (**B**) Western blot showing cleavage of pro-PL-MMP14 by the purified SARS-CoV-2 PL^pro^. (**C**) Gelatin zymography assay to confirm whether cleaved MMP14 is active using pro-MMP2.

## Data Availability

All data are presented within the manuscript.

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
