# Peer review of "Selective Degradation and Inhibition of SARS-CoV-2 3CLpro by MMP14 Reveals a Novel Strategy for COVID-19 Therapeutics"

_ijms, 2025, doi:10.3390/ijms26199401_

Round 1
Reviewer 1 Report (New Reviewer)
Comments and Suggestions for Authors
Comments on ijms-3870027
I have read the manuscript. I think the idea of ​​a new pro-PL-MMP14 construct designed for PLpro activation is particularly novel, because it suggests a more targeted antiviral strategy that could minimize side effects.
I suggest only some minor modifications.
I suggest to expand the literature background in the introduction:
- There is currently a lack of previous attempts or examples that focus on viral enzyme degradation via endogenous enzymes (e.g. MMPs). Please provide additional literature context.
- On what basis was the target protein selected?
- How do other strategies (e.g. drug repurposing or alternative mechanisms) fit into this approach?
- Please also present results on other therapeutic strategies in a few sentences, for example, what examples of other methods based on different mechanisms are provided and how they differ from MMP14-based protein degradation. (I suggest the following study: https://biomedres.us/fulltexts/BJSTR.MS.ID.006392.php)
- The clear hypothesis is missing from the end of the introduction, which is based on previous knowledge, and it would be good if the authors explained the method used to verify their hypothesis.
Results:
- It would be good to somehow investigate the real infection cycle of SARS-CoV-2. If this is not possible, it would be good to at least mention among the limitations of the study that experiments on real viruses need to confirm the hypothesis.
- The figure3 is more of a table-like, text-based format; I suggest a graphic representation that would visually aid the reader in interpretation.
- Figure 6: indicate the replicate number and add error bars. Highlight more visually that MMP14 does not cleave the substrate
Discussion:
- Essentially, MMP14 is a membrane-bound protease, therfore the question arises: how does it access cytosolic 3CLpro? This question should be explored in detail in the discussion section.
Title:
- The title is too general; I suggest including the following informations as well. Selective degradation, enzyme inhibition, and that it also reports on a new, recombinant construct.
Author Response
Please see the attachment

Reviewer 2 Report (New Reviewer)
Comments and Suggestions for Authors
This study focuses on the degradation effect of MMP14 on the key protease 3CLpro of SARS-CoV-2 and its antiviral potential. It has clear innovation, reasonable experimental design, and sufficient workload, and overall meets the requirements for journal publication.
- Issue regarding internal reference calibration for quantifying 3CLpro degradation efficiency.
The article states that "at least 75% of intact 3CLpro is degraded by MMP14 within 15 minutes" based on Odyssey densitometry analysis. However, it does not clarify whether an internal reference protein (such as MMP14 in the lanes or a loading control) was used for calibration in this quantification process, nor does it confirm whether the loading amounts in different lanes were completely consistent (e.g., verified by the total protein amount via Coomassie blue staining). If there are differences in loading amounts or no calibration was performed, it may lead to misjudgment of the degradation efficiency. - Issues with the format specifications of the Edman sequencing result charts. In Figure 3A, the "Amino Acid" column in the "Edman degradation" table does not specify whether it refers to "the main amino acid detected in each cycle (e.g., S is the main one, V is a trace impurity)" or "mixed signals caused by detection errors". The chart notes also do not explain the meaning of this format, which is likely to cause misunderstandings among readers.
- Insufficient discussion on the mechanism of cat-MMP14 entering the cytoplasm. The article mentions that "MMP14 is typically a membrane-bound protease (on the cell surface), and 3CLpro is located in the cytoplasm" and speculates that cat-MMP14 may interact with 3CLpro in the cytoplasm. However, it does not further analyze the specific mechanism by which cat-MMP14 enters the cytoplasm (such as whether through endocytosis, abnormal endoplasmic reticulum-Golgi transport, or direct transmembrane diffusion). The lack of this mechanism will affect the complete understanding of the MMP14 antiviral pathway.
Author Response
Please see the attachment

This manuscript is a resubmission of an earlier submission. The following is a list of the peer review reports and author responses from that submission.
Round 1
Reviewer 1 Report
Comments and Suggestions for Authors
The authors Lee et. al describe describe cleavage of SARS-CoV-2 3CLpro by MMP14. They showed that MMP14 is the only among other 5 MMPs tested to cleave the 3CLpro in vitro and they showed that over expression of MMP14 in HEK293 with virus infected cells reduces the pseudovirus replication.
Before I can support publication of this work the authors must address the issues exposed below.
Major remarks:
As mentioned in introduction, MMP14 is involved in a number of physiological processes therefore their suggestion to use MMP14 for inactivation of 3CLpro for treatment of SARS-CoV-2 infections is not sound. To make the suggestion sound, the authors should point out how these other processes will not be effected, which largely accedes the scope of this manuscripts. Hence, the authors should modify the manuscript accordingly.
To make a sound contribution to science, I suggest the authors to explore whether MMP14 plays a role in defense against the SARS-CoV-2 virus. A cell knockout of MMP14 in cell culture may reveal whether this is indeed so.
Minor remark.
Lines 84-85. “We detected 16 potential MMP14 cleavage sites on the SARS-CoV-2 3CLpro.” and
Lines 234-236. “Based on our extensive experience with MMP14, we searched all known MMP14 cleavage sites in SARS-CoV-2 3CLpro and PLpro and found eighteen and two potential cleavage sites on 3CLpro and PLpro, respectively.” The authors should provide their list in the manuscript. Moreover, the quote from introduction indicates, that these cleavages were discovered within the frame of this work, hence experimental details of their discovery should be included in this manuscript. Besides, the numbers of cleavages are different in these two different quotes.
Author Response
Major remarks:
As mentioned in introduction, MMP14 is involved in a number of physiological processes therefore their suggestion to use MMP14 for inactivation of 3CLpro for treatment of SARS-CoV-2 infections is not sound. To make the suggestion sound, the authors should point out how these other processes will not be effected, which largely accedes the scope of this manuscripts. Hence, the authors should modify the manuscript accordingly.
To make a sound contribution to science, I suggest the authors to explore whether MMP14 plays a role in defense against the SARS-CoV-2 virus. A cell knockout of MMP14 in cell culture may reveal whether this is indeed so.
Authors’ answer: We completely agree with the reviewer’s concern about MMP14 being involved with various physiological processes that we do not want to interfere with. We actually thought about this very issue right from the beginning and came up with an idea to avoid it. The pro-domain intact MMP14 has been known to be inactive, and furin transiently binds to the pro-form of MMP14 (pro-MMP14) and cleaves the pro-domain sequence of pro-MMP14 to yield an active form of MMP14 (act-MMP14). Our newly designed recombinant construct, pro-PL-MMP14, contains a SARS-CoV-2 PLpro (another SARS-CoV protease) cleavage site (LKGGA) between the pro-domain and act-MMP14 with a C-terminal His6-tag, and Arg108 that has been mutated to Pro to avoid cleavage by furin (Figure 1a). We were able to purify this pro-PL-MMP14 protein and verify by SDS-PAGE and western blotting analyses against two antibodies, anti-His and anti-MMP14. To investigate cleavage of pro-PL-MMP14 by SARS-CoV-2 PLpro, we also purified SARS-CoV-2 PLpro enzyme. An act-MMP14 (cleaved form) was produced in the presence of PLpro (Figure 1b, blue box). Act-MMP2 was generated only when both pro-PL-MMP14 and PLpro were present, indicating activation of pro-PL-MMP14 by SARS-CoV-2 PLpro (Figure 1c). Therefore, this modified recombinant pro-PL-MMP14 will be used to inhibit viral replication for further development of therapeutic intervention with hopefully moderate to no adversary effect on uninfected cells since pro-PL-MMP14 will be activated in only SARS-CoV infected cells. We are in process of writing a follow-up manuscript with below results and other results, and hence we are hoping not to include these data into the current manuscript.
Minor remark.
Lines 84-85. “We detected 16 potential MMP14 cleavage sites on the SARS-CoV-2 3CLpro.” and
Lines 234-236. “Based on our extensive experience with MMP14, we searched all known MMP14 cleavage sites in SARS-CoV-2 3CLpro and PLpro and found eighteen and two potential cleavage sites on 3CLpro and PLpro, respectively.” The authors should provide their list in the manuscript. Moreover, the quote from introduction indicates, that these cleavages were discovered within the frame of this work, hence experimental details of their discovery should be included in this manuscript. Besides, the numbers of cleavages are different in these two different quotes.
Authors’ answer: We apologize for our mistake. Thank you so much for noticing this. We added protein sequence of the SARS-CoV-2 3CLpro and a total of 17 potential cleavage sites by MMP14 to Figure 2A. We have revised both sixteen and eighteen to seventeen in the revised manuscript. The new Figure 2 is shown here.

Reviewer 2 Report
Comments and Suggestions for Authors
The paper by Lee et al addresses the role of MMP14 as a potential inhibitor of 3CPpro a key enzyme of the SRAS-CoV-2 virus. The main findings are that the catalytic activity of MMP14 can degrade 3CLpro and thus impair SARS-CoV-2 replication in cells. The purified catalytic domain seems more efficient and may have less side effects than full-length MMP14.
The claims rely on biochemical degradation and binding assays in "test tubes" and some experiments in cell lines.
Some improvements are necessary:
1/ Most experiments appear to have been done only once. No reference to sample size (n?) and whether there were any technical or exprimental replicates made. It should be clearly mentioned how many times an experiments has been made. If there is more than one, there should a clear explanation about quantification and what error bars mean. Whether the selected image for the figure is a representative example out of X samples etc... the other images (for instance other gels for the technical/exprimental replicates) should be provided uncropped and unmodified along side the one selected for the main figure into a supp data file.
2/ The comparison between the effect of cat-MMP14 and full-length-MMP14 in cells is difficult to interpret. In the methods it is said that both the cat domain and Fl-mmp14 were cloned in a vector and cells were transfected. One assumes FL-MMP14 contains a signal peptide driving expression/translation of MMP14 through the secretion pathway (ER/Golgi) whereas the cat domain only has its own sequence, thus expression/translation should occur in the cytosol. Therefore, if interaction between MMP14 and 3CLpro needs to take place in the cytosol it is understandable that the cat domain will reach efficiency before the FL version of MMP14 since only a fraction may end up in the cytosol while most of it should be on the cell surface. It would be important to compare similar situations: either a deltaSP version of FL MMP14 vs cat-domain only, or the cat domain of MMP14 in a vector containing the SP to drive secretion vs FL-MMP14 and see if they affect cleavage of 3CPpro similarly.
3/ The claim that cat-MMP14 will have less side effect than MMP14 given its inability to activate mmp2 should be either removed or toned down. Overexpressing a catalytic domain may have dominant-negative effect. In addition, not all functions of MMP14 (or MMPs in general) rely on its/their catalytic activity and a lower efficiency at cleaving one known substrate do not preclude massive side effects related to other aspects of MMP14's biology.
4/ In Figure 2B why the 12Kd fragment of cleaved 3CLpro progressively disappear instead of accumulating over time?
5/ How can the authors know that the 12kDa fragment is a degradation product of 3CLpro and not a fragment due to MMP14 autocatalysis? This fragment should be purified and sent to mass spec for identification.
6/ Figure 3A, the whole gel should be provided uncropped and unmodified.
7/ Experiments in Figure 3B involve other MMPs (namely 2, 7, 9 and 13) which apparently did not cleave 3CLpro. First, there should be some positive controls of their respective efficiency to check that they were indeed functional under these conditions. Also, under MMP2 and MMP13, there is a 12kDa fragment similar to the one observed after treatment with MMP14. Why are these fragments not considered a product of 3CLpro degradation? From the interpretation of the data on figure 2A and B it would appear that mmp2 and 13 do cleave 3CLpro (fig 3B). Or as mentioned above these could be autocatalysis fragments from the MMPs themselves.
In addition, if confirmed that these other enzymes cannot cleave 3CLpro it does not mean that no other MMPs will not also do it so the claims that only MMP14 can cleave 3CLpro should be moderated. .
8/ An attempt at validating physical interaction between MMP14 and 3CLpro in cells would be much welcome. Double immunostaining? Colocalization assay such as proximity ligation assays or proximity biotinylation assay? Co-IP from cell extracts?
Author Response
Reviewer 2.
The paper by Lee et al addresses the role of MMP14 as a potential inhibitor of 3CLpro a key enzyme of the SRAS-CoV-2 virus. The main findings are that the catalytic activity of MMP14 can degrade 3CLpro and thus impair SARS-CoV-2 replication in cells. The purified catalytic domain seems more efficient and may have less side effects than full-length MMP14.
The claims rely on biochemical degradation and binding assays in "test tubes" and some experiments in cell lines.
Some improvements are necessary:
1/ Most experiments appear to have been done only once. No reference to sample size (n?) and whether there were any technical or exprimental replicates made. It should be clearly mentioned how many times an experiments has been made. If there is more than one, there should a clear explanation about quantification and what error bars mean. Whether the selected image for the figure is a representative example out of X samples etc... the other images (for instance other gels for the technical/exprimental replicates) should be provided uncropped and unmodified along side the one selected for the main figure into a supp data file.
Authors’ answer: We performed in vitro proteolysis experiments twice for figures 2 and 3. All gel images were replaced with full size uncropped/unmodified ones as suggested by the reviewer. SPR direct binding analyses, fluorescent-based enzymatic assays, and pseudovirus assays were repeated three times. All error bars reported in this manuscript indicate the standard deviation from triplicates (n=3). We added these explanations in method sections and figure legends.
2/ The comparison between the effect of cat-MMP14 and full-length-MMP14 in cells is difficult to interpret. In the methods it is said that both the cat domain and Fl-mmp14 were cloned in a vector and cells were transfected. One assumes FL-MMP14 contains a signal peptide driving expression/translation of MMP14 through the secretion pathway (ER/Golgi) whereas the cat domain only has its own sequence, thus expression/translation should occur in the cytosol. Therefore, if interaction between MMP14 and 3CLpro needs to take place in the cytosol it is understandable that the cat domain will reach efficiency before the FL version of MMP14 since only a fraction may end up in the cytosol while most of it should be on the cell surface. It would be important to compare similar situations: either a deltaSP version of FL MMP14 vs cat-domain only, or the cat domain of MMP14 in a vector containing the SP to drive secretion vs FL-MMP14 and see if they affect cleavage of 3CPpro similarly.
Authors’ answer: We thank the reviewer for this insightful advice even though we cannot finish it during permitted revision deadline, which is 10 days. We will surely do it in near future as suggested. We agree that cat-MMP14 have a better chance to interact with 3CLpro in cytosol than FL-MMP14. As described above in answer to the reviewer 1, we engineered a novel pro-form of MMP14 (pro-PL-MMP14) that can be selectively activated only by SARS-CoV-2 PLpro enzyme, another essential SARS-CoV protease.
3/ The claim that cat-MMP14 will have less side effect than MMP14 given its inability to activate mmp2 should be either removed or toned down. Overexpressing a catalytic domain may have dominant-negative effect. In addition, not all functions of MMP14 (or MMPs in general) rely on its/their catalytic activity and a lower efficiency at cleaving one known substrate do not preclude massive side effects related to other aspects of MMP14's biology.
Author’s answer: We have removed the claim that cat-MMP14 will have less side effect than MMP14 in both result and discussion sections as suggested. We fully agree on using simple cat-MMP14 as a therapy could be associated with other adversary effects. Hence, we already explored an option we explained in answer to the reviewer 1. The pro-domain intact MMP14 has been known to be inactive, and furin transiently binds to the pro-form of MMP14 (pro-MMP14) and cleaves the pro-domain sequence of pro-MMP14 to yield an active form of MMP14 (act-MMP14). Our newly designed recombinant construct, pro-PL-MMP14, contains a SARS-CoV-2 PLpro (another SARS-CoV protease) cleavage site (LKGGA) between the pro-domain and act-MMP14 with a C-terminal His6-tag, and Arg108 that has been mutated to Pro to avoid cleavage by furin (Figure 1a). We were able to purify this pro-PL-MMP14 protein and verify by SDS-PAGE and western blotting analyses against two antibodies, anti-His and anti-MMP14. To investigate cleavage of pro-PL-MMP14 by SARS-CoV-2 PLpro, we also purified SARS-CoV-2 PLpro enzyme. An act-MMP14 (cleaved form) was produced in the presence of PLpro (Figure 1b, blue box). Act-MMP2 was generated only when both pro-PL-MMP14 and PLpro were present, indicating activation of pro-PL-MMP14 by SARS-CoV-2 PLpro (Figure 1c). Therefore, this modified recombinant pro-PL-MMP14 will be used to inhibit viral replication for further development of therapeutic intervention with hopefully moderate to no adversary effect on uninfected cells since pro-PL-MMP14 will be activated in only SARS-CoV infected cells.
4/ In Figure 2B why the 12Kd fragment of cleaved 3CLpro progressively disappear instead of accumulating over time?
5/ How can the authors know that the 12kDa fragment is a degradation product of 3CLpro and not a fragment due to MMP14 autocatalysis? This fragment should be purified and sent to mass spec for identification.
Author’s answer for 4 and 5: Thanks for pointing this out. As can be seen in Figure 2C (new Figure), 3CLpro cleavage by MMP14 happens fairly fast within the first 15 min, producing a fragment at ~12kDa. We agree that this fragment could be generated from either/both 3CLpro or/and MMP14. We suspect that this fragment continues to be degraded as time goes by, which is why the band at 12 kDa is being fade away instead of getting darker. As the reviewer mentioned, we agree that there is a possibility of autocatalysis of MMP14 since MMP14 band without GM6001 is weaker than that of MMP14 in the presence of GM6001 in Figure 3A. Hence, we will identify the origin of the 12 kDa fragment by mass spec as suggested. We have changed the Figure 2 fragment label from “3CLpro fragment” to “cleaved fragment”.
6/ Figure 3A, the whole gel should be provided uncropped and unmodified.
Author’s answer: We have replaced figure 3A with the whole uncropped and unmodified gel picture.
7/ Experiments in Figure 3B involve other MMPs (namely 2, 7, 9 and 13) which apparently did not cleave 3CLpro. First, there should be some positive controls of their respective efficiency to check that they were indeed functional under these conditions. Also, under MMP2 and MMP13, there is a 12kDa fragment similar to the one observed after treatment with MMP14. Why are these fragments not considered a product of 3CLpro degradation? From the interpretation of the data on figure 2A and B it would appear that mmp2 and 13 do cleave 3CLpro (fig 3B). Or as mentioned above these could be autocatalysis fragments from the MMPs themselves.
In addition, if confirmed that these other enzymes cannot cleave 3CLpro it does not mean that no other MMPs will not also do it so the claims that only MMP14 can cleave 3CLpro should be moderated. .
Author’s answer: We toned down and moderated as suggested by the reviewer. We confirmed enzyme assay buffers for these tested MMPs share similar composition and also confirmed that they are active. Around 12 kDa fragments produced by MMP2 and MMP13 will also be identified by mass spec in near future. We have removed “only MMP14 can cleave”.
8/ An attempt at validating physical interaction between MMP14 and 3CLpro in cells would be much welcome. Double immunostaining? Colocalization assay such as proximity ligation assays or proximity biotinylation assay? Co-IP from cell extracts?
Author’s answer: Thanks for the great suggestions! For this manuscript, we are hoping to establish physical interaction by SPR direct binding analysis. We do agree the need of physical interaction in the cells as well. As shown in Figure 1 in answer to the reviewer 1 above, we already made a pro-form of MMP14 (pro-PL-MMP14). We will test in cell physical interaction in HEK293 cells after virus infection using immunostaining and Co-IP, which we are hoping to put in our follow-up manuscript.

Round 2
Reviewer 1 Report
Comments and Suggestions for Authors
add 1). The suggestion to introduce a PLpro cleavage site sequence in the MMP14 is an interesting idea, however, it only supports this manuscript when included.
add 2) The source of cleavages mentioned in figure 2A has not been revealed. While reading revised ms, remarks of reviewer 2 and the author responses it became clear to me that these cleavages are actually only putative. Without their sequence experimentally determined bands in the gels can originate from any protein included in the assay.
To gain my support for publication of this ms
- the authors must include the experiments mentioned in response add 1) in the ms.
- the potential cleavage sites do not corroborate anything. Experimental determination of sequences of the bands on the gels is mandatory to support the conclusions.
Author Response
RE2: Manuscript ID: ijms-1484661
Title: MMP14 cleavage of SARS-CoV-2 3CLpro and implications for drug discovery against SARS-CoV-2
Reviewer 1.
add 1) The suggestion to introduce a PLpro cleavage site sequence in the MMP14 is an interesting idea, however, it only supports this manuscript when included.
Authors’ answer: We added it as Figure 6D as suggested by the reviewer. New Figure 6 is shown below.
add 2) The source of cleavages mentioned in figure 2A has not been revealed. While reading revised ms, remarks of reviewer 2 and the author responses it became clear to me that these cleavages are actually only putative. Without their sequence experimentally determined bands in the gels can originate from any protein included in the assay.
To gain my support for publication of this ms. the authors must include the experiments mentioned in response add 1) in the ms.
the potential cleavage sites do not corroborate anything. Experimental determination of sequences of the bands on the gels is mandatory to support the conclusions.
Authors’ answer: It seems like our word “detected” might have been misleading. We have simply searched 8 reported MMP14 cleavage sequences on SARS-CoV-2 3CLpro and found 17 potential cleavage sites. We revised line 85 from “detected” to “ have searched these 8 cleavage sequences and found” in order to avoid confusion. It seems like the reviewer focused on the cleaved fragments and its origin. However, our main focus is actually 3CLpro being disappeared. The data we presented in this manuscript clearly support that MMP14 directly binds and cleaves 3CLpro, inhibit 3CLpro enzyme activity in vitro, and inhibit SARS-CoV-2 Pseudovirus replication in 293T cells. We wonder whether finding out where exactly MMP14 cleaves on 3CLpro would provide any additional information since our focus and goal is to get rid of 3CLpro in SARS-CoV infected cells. Our hypothesis is that MMP14 cleaves 3CLpro on multiple locations, and it doesn’t matter where as long as this cleavage event disable the function of the 3CLpro enzyme, resulting in inhibition of virus replication only in infected cells. Thank you.
